# PRIVACY AUDITING OF MACHINE LEARNING USING MEMBERSHIP INFERENCE ATTACKS

## ABSTRACT

Membership inference attacks are used as an auditing tool to quantify the private information that a model leaks about the individual data points in its training set. Membership inference attacks are influenced with different uncertainties that an attacker has to resolve about training data, training algorithm, and the underlying data distribution. Thus attack success rates, of many attacks in the literature, do not precisely capture the information leakage of models about their data, as they also reflect other uncertainties that the attack algorithm has. In this paper, we present a hypothesis testing framework that can explain the implicit assumptions and also the simplifications made in the prior work. We also derive new attack algorithms from our framework that can achieve a high AUC score while also highlighting the different factors that affect their performance. Our algorithms capture a very precise approximation of privacy loss in models, and can be used as a tool to perform an accurate and informed estimation of privacy risk in machine learning models. We provide a thorough empirical evaluation of our attack strategies on various machine learning tasks trained on benchmark datasets.

## 1 INTRODUCTION

Machine learning systems have come under intense scrutiny of the regulatory authorities in the past few years. Veale et al. (2018) argue that machine learning models could be considered personal data due to their susceptibility to inference attacks that can recover sensitive information about training data just from the models. Membership inference attacks (Homer et al., 2008; Dwork et al., 2015; Shokri et al., 2017) and reconstruction attacks (Dinur & Nissim, 2003; Song et al., 2017; Carlini et al., 2020) are the main inference attacks that highlight, and can quantify, the privacy risk of releasing aggregate information computed on sensitive data (Dwork et al., 2017). The focus of this paper will be on membership inference attacks for measuring privacy risk. Organizations such as the ICO (UK) and NIST (US) have highlighted membership inference as a potential confidentiality violation and privacy threat to the training data (Murakonda & Shokri, 2020). This has lead to the development of open-source tools [1] and capabilities in widely-used ML libraries [2] for measuring privacy risk from machine learning models using membership inference attacks.

Although the approach of quantifying privacy risk through membership inference attacks is gaining traction, the attack success, as measured by a lot of works, cannot be completely attributed to information leakage from the models and hence their privacy risk. Various factors such as the distribution of training data, difference in distributions of the train and test data may provide an over-estimate or under-estimate of the actual privacy risk from the model (Erlingsson et al., 2019; Humphries et al., 2020). Theoretical analyses that connect the success of membership inference to privacy risk through the framework of differential privacy avoid this issue by slightly modifying how the attack performance is measured (Yeom et al., 2018; Jagielski et al., 2020; Nasr et al., 2021; Malek et al., 2021). Instead of measuring the leakage from a particular model, these works aim at measuring the worst-case leakage of the training algorithm (for a given model architecture), and construct multiple models with and without one training point and keep the rest of training set fixed. The performance of the attack (false positive and false negative errors) is then computed over these sampled models.

---

[1] https://github.com/privacytrustlab/ml_privacy_meter
[2] https://blog.tensorflow.org/2020/06/introducing-new-privacy-testing-library.html

However, in practice and during auditing, the performance needs to be computed based on the points in the training set versus the population data (test set) for a given fixed model. Various works rely on measuring privacy risk through the performance of membership inference attacks, but the subtle differences in how the attack is formulated mean that they might associate different causes of attack success to privacy loss. When auditing machine learning models, we need to pay attention to what exactly we are measuring and how it relates to the information leakage of the machine learning algorithm and not other factors such as the prior knowledge of the attacker.

Previous membership inference attacks are designed for high *overall* performance, i.e. to succeed for *most* member (or non-member) data points of *most* target models. With few exception, they train shadow (reference) models on datasets randomly sampled from a population, to mimic the *general behavior* of models on member (or non-member) data points. This general behavior, however, does not capture the behavior of specific models on specific samples. As a result, the limited performance of shadow model attack does not precisely measure the model (sample)-specific information leakage.

In this work, we focus on designing membership inference attacks that more precisely measure what a target model leaks about each individual training data, in a binary hypothesis testing framework. We start from the (constant) loss threshold attack, and study how to further design different attack strategy (loss threshold) for different target model (sample). We derive multiple attack strategies (existing and new) from this hypothesis test formulation via different approximations for the null hypothesis. We also formulate the performance of an attack inside the hypothesis testing framework, via the trade-off between its false positive rate and true positive rate. Following the methodology, we not only derive and explain existing shadow (reference) model attacks (Attack S and R), but also design new attacks (Attack P and D) that offer more accurate privacy auditing for machine learning models, through model-dependent and (or) sample-dependent attack strategies. We empirically evaluate and compare the attack performance (TPR-FPR curve and AUC score) and computation cost of our new attacks (notably Attack D), and the prior attacks on multiple datasets.

## 2 ATTACK FRAMEWORK

Our objective is to design a framework that enables auditing the privacy loss of machine learning models in the black-box setting (where only model outputs —and not their parameters or internal computations— are observable). This framework needs to have three elements: (i) the inference game as the evaluation setup; (ii) the indistinguishably metric to measure the privacy risk, and (iii) the construction of membership inference attack as hypothesis testing. The notion of privacy underlying our framework is primarily based on differential privacy, and multiple pieces of this framework are generalizations of existing inference attacks against machine learning algorithms. Instead of focusing on designing new attacks, we present the important design choices for constructing and evaluating membership inference attacks, for the purpose of having a precise privacy auditing. We identify different sources of uncertainty that influence the error of inference attacks, and can lead to miscalculation of the privacy loss of a model.

We quantify privacy loss of a model in a hypothetical **inference game** between a challenger and an adversary. We are given a private training set $D$, and a model $\theta$ which is trained on $D$ using a training algorithm $\mathcal{T}$. The challenger samples a random data point $(x_1, y_1)$ from the training set (a member), and a random data point $(x_0, y_0)$ from the data *population* outside the training set (a non-member). He then randomly selects one of the two (member or non-member) $b \sim \{0, 1\}$ with probability $\frac{1}{2}$, and shares the selected data point $(x_b, y_b)$ and the model's output $f(x_b; \theta)$ with the adversary. The adversary's task is to determine if the data point is a member or not (i.e., guess $b$).[3]

---

[3]We need to emphasize that there is a difference between the privacy loss of a machine learning *algorithm* and that of the specific *models* trained with the algorithm. A model is a given instance of a training algorithm, thus its leakage needs to be computed with respect to the individual data records in its specific training set. This subsequently means that the privacy loss of an algorithm varies depending on the randomness in sampling of its training data, and the randomness of the training algorithm. One can analyze the privacy loss of an algorithm as, for example, its worst case privacy loss (as in differential privacy). Differentially private algorithms enforce an upper bound on the privacy loss of an algorithm over all models with respect to all possible training data. Thus, in the inference game for differential privacy, the privacy loss of the training *algorithms*, would be the worst case privacy loss over all choices of $D$, $(x_0, y_0)$, and $(x_1, y_1)$ in our inference game for model privacy.

We use an **indistinguishability** measure (which is the basis of differential privacy) to define privacy of individual training data of a model. According to this measure, the *privacy loss* of the model with respect to its training data is the adversary's success in distinguishing between the two possibilities $b = 0$ vs $b = 1$ over multiple repetitions of the inference game. Naturally, the inference attack is a **hypothesis test**, and adversary's error is composed of the false positive (i.e., inferring a non-member as member) and false negative of the test. In practice, the error of adversary in each round of the inference game depends of multiple factors:

- The true leakage of the model about the target data $(x_b, y_b)$ when $b = 1$.

- The uncertainty (belief or background knowledge) of attack algorithm about the population data

- The adversary's uncertainty about the training algorithm $\mathcal{T}$

- The uncertainty about all training data except the target data $(x_b, y_b)$

- The attack dependency on the target data $(x_b, y_b)$, and the model $\theta$

In the ideal setting, we only want the attack error to be dependent on the true leakage of the model about the target data (i.e., whether the same model trained with and without $(x_b, y_b)$ are distinguishable from each other). To this end, and to cancel out the effect of other uncertainties, an attack algorithm and the evaluation setup for the inference game need to be designed based on the following principle: The population data used for constructing the attack algorithm, and evaluating the inference game, need to be similar, in distribution, to the training data. This is to minimize the impact of prior belief (what could have been sampled for the training set) in the performance of the inference attack. This is not hard to achieve as all the process (of constructing the hypothesis testing attack, and evaluating it) is controlled by the auditor. By violating this principle, we might overestimate the privacy loss (by making the test dependent on a distinct prior knowledge) or underestimate the privacy loss (by evaluating the inference attack on a population data distribution for which it was not constructed).

Another crucial requirement is that the privacy audit needs to output a detailed report, which captures the uncertainty of the attack. Reporting one number as the accuracy of the attack, as it is mostly reported in the literature, is not an informative report. Given the attack being a hypothesis test, the audit report needs to include the analysis of the error versus power of the test: if we can tolerate a certain level of false positive rate in the inference attack, how much would be the true positive rate of the attack, over the random samples from member and non-member data? The area under the curve for such an analysis reflects the chance that the membership of a random data point from the population or the training set can be inferred correctly.

## 3 CONSTRUCTING MEMBERSHIP INFERENCE ATTACKS

The adversary in the membership inference game can observe the output of a target machine learning model $\theta$, trained on unknown dataset $D$. He also gets a precise target data point $z$ as input, and is expected to output 0 or 1 to guess whether the sample $z$ is in the dataset $D$ or not. We use likelihood ratio test (LRT) as the most powerful criterion for choosing among membership hypotheses, under the following assumptions.

1. The adversary knows, and can sample from the underlying data distribution $\pi(z)$ over population $z = (x_z, y_z) \in D_{pop}$.
2. A randomized training algorithm $\mathcal{T} : D \mapsto \theta$ takes in a training dataset $D$, which consists of i.i.d. samples from the data population, and produces a model $\theta$ that incurs a low loss $\sum_{(x,y) \in D} \ell(\theta, x, y)$ on dataset $D$. We denote $P(\theta|D)$ to be the posterior distribution of trained model $\theta$ given training dataset $D$.

**Definition 3.1 (Approximated LRT for Membership Inference)** *Let $(\theta, z)$ be random samples from the joint distribution of target model and target data point, specified by one of the following membership hypotheses.*

$$H_0 : D \xleftarrow{n\ i.i.d.samples \sim \pi(z)} D_{pop}, \theta \sim \mathcal{T}(D), z \xleftarrow{sample \sim \pi(z)} D_{pop} \tag{1}$$

$$H_1 : D \xleftarrow{n\ i.i.d.samples \sim \pi(z)} D_{pop}, \theta \sim \mathcal{T}(D), z \xleftarrow{sample} D \tag{2}$$

The likelihoods function of hypothesis $H_0$ and $H_1$, given observed target model $\theta$ and target data point $z$, is as follows (detailed derivations are in the Appendix).

$$L(H_0|\theta, z) = P_{H_0}(\theta, z) = \pi(z) \cdot \mathbf{E}_{D \sim \pi^n}[P(\theta|D)] \tag{3}$$

$$L(H_1|\theta, z) = P_{H_1}(\theta, z) = \pi(z) \cdot \mathbf{E}_{D' \sim \pi^{n-1}}[P(\theta|D' \cup z)] \tag{4}$$

Now we follow the previous construction of Bayes-optimal membership inference attack Sablayrolles et al. (2019), and model the posterior distribution $P(\theta|D)$ of trained model as follows.

$$P(\theta|D) = \frac{e^{-\frac{1}{T} \sum_{(x,y) \in D} \ell(\theta, x, y)}}{\int e^{-\frac{1}{T} \sum_{(x,y) \in D} \ell(\theta, x, y)} d\theta}, \tag{5}$$

where $T$ is a temperature constant that allows some randomness in the training algorithm $\mathcal{T}$. The equation 5 holds for Bayesian learning algorithms, such as stochastic gradient descent Polyak & Juditsky (1992), deterministic MAP (Maximum A Posteriori) inference (for $T \to 0$), and Bayesian posterior sampling Welling & Teh (2011) (for $T = 1$). Therefore, the LRT statistics can be computed as follows (detailed derivations are in the Appendix).

$$LR(\theta, z) = \frac{L(H_0|\theta, z)}{L(H_1|\theta, z)} = \frac{\mathbf{E}_{D \sim \pi^n}[P(\theta|D)]}{\mathbf{E}_{D' \sim \pi^{n-1}}[P(\theta|D' \cup z)]} \tag{6}$$

$$\approx e^{\frac{1}{T} \ell(\theta, x_z, y_z)}, \tag{7}$$

The LRT hypothesis test rejects $H_0$ when the LRT statistic is small. By equation 7, the rejection region $\{(\theta, z) : LR(\theta, z) \leq c\}$ can be approximated as follows.

$$\left\{ (\theta, z) : \ell(\theta, x_z, y_z) \leq T \cdot \log c \right\} \tag{8}$$

The above approximated LRT strategy compares a constant threshold $T \cdot \log c$ with the loss $\ell(\theta, x_z, y_z)$ of the target model $\theta$ on the target data point $z$. This recovers the commonly used (constant) loss threshold attacks in literature Nasr et al. (2019); Sablayrolles et al. (2019). However, by derivations in the appendix (equation 48), a more accurate LRT attack strategy would compare the loss $\ell(\theta, x_z, y_z)$ with a threshold function $T \cdot \log \left( \frac{c \cdot \mathbf{E}_{D' \sim \pi^{n-1}}[P(\theta|D')]}{\mathbf{E}_{D \sim \pi^n}[P(\theta|D)]} \right)$ that depends on the target model $\theta$. Therefore, the attack in equation 8 with constant threshold $c_\alpha$ can overly simplify the LRT, thus limiting its performance. This motivates our design of attacks with model-dependent and sample-dependent thresholds, with the objective of improving the attack performance.

**Our general template for attack construction.** Building on the approximate LRT equation 8, we derive the following variant of sample-dependent and (or) model dependent attack strategy.

$$\text{If } \ell(\theta, x_z, y_z) \leq c_\alpha(\theta, x_z, y_z), \text{ reject } H_0, \tag{9}$$

where $c_\alpha(\theta, x_z, y_z)$ is a threshold function chosen by the attacker under $\alpha$ tolerance of false positive rate (FPR), i.e., $c_\alpha(\theta, x_z, y_z)$ satisfies the following equation which controls false positive rate.

$$\mathbf{E}_{P_{H_0}(\theta, x_z, y_z)} \left[ \mathbf{1}_{\ell(\theta, x_z, y_z) \leq c_\alpha} \right] = \alpha \tag{10}$$

In the following attacks, we approximate the joint distribution $P_{H_0}(\theta, z)$ with empirical distribution over its samples. This facilitates solving equation 10, which give valid attack threshold $c_\alpha(\theta, x_z, y_z)$.

## 3.1 ATTACK S: MIA VIA S̲HADOW MODELS

We first formalize a shadow model membership inference attack S based on Shokri et al. (2017), that effectively uses label-dependent attack threshold $c_\alpha(y_z)$, as follows.

$$\text{If } \ell(\theta, x_z, y_z) \leq c_\alpha(y_z), \text{ reject } H_0, \tag{11}$$

where the threshold function $c_\alpha(y_z)$ satisfies equation 10 and ensures false positive rate $\alpha$. To approximate the joint distribution $P_{H_0}(\theta, x_z, y_z)$ in equation 3, the attacker samples the following set $S$ of shadow models and shadow data points from the joint distribution $P_{H_0}(\theta, z)$.

$$S = \cup_{i=1,2,\cdots} \left\{ (\theta_i, z_1^i), (\theta_i, z_2^i), \cdots \right\} \tag{12}$$

$$\forall i = 1, 2, \cdots, \quad \theta_i \sim \mathcal{T}(D_i), D_i \xleftarrow{n \ i.i.d. samples \sim \pi(z)} D_{pop} \tag{13}$$

$$\forall i = 1, 2, \cdots, \quad z_1^i, z_2^i, \cdots i.i.d. \sim \pi(z) \tag{14}$$

The shadow models $\theta_s = \{\theta_i\}_{i=1,2,...}$ trained on population datasets approximate the distribution $\mathbf{E}_{D \sim \pi^n}[P(\theta|D)]$ in equation 3, and the shadow points $z_1^i, z_2^i, \cdots$ for the $i$-th shadow model approximate the population data distribution $\pi(z)$ in equation 3. Therefore, we approximate the joint distribution $P_{H_0}(\theta, z)$ in equation 10 with the empirical distribution over set $S$, as follows.

$$\frac{|\{(\theta, z) \in S : \ell(\theta, x_z, y_z) \leq c_\alpha(y_z)\}|}{|S|} = \alpha \tag{15}$$

One sufficient condition for equation 15 to hold is that for any fixed data label $y \in Y$, we have

$$\frac{|\{(\theta, z) \in S : \ell(\theta, x_z, y_z) \leq c_\alpha(y_z) \text{ and } y_z = y\}|}{|\{(\theta, z) \in S : y_z = y\}|} = \alpha \tag{16}$$

By solving equation 16 for every $y \in Y$, we compute that $c_\alpha(y)$ equals the the $\alpha$-percentile for the histogram of loss values $\ell(\theta, x_z, y_z)$ over samples $\{(\theta, z) \in S : y_z = y\}$. This recovers the class-dependent attack threshold function $c_\alpha(y_z)$ that we use in Attack S.

### 3.2 ATTACK P: MODEL-DEPENDENT MIA VIA POPULATION DATA

We design a new membership inference Attack P that uses attack threshold function $c_\alpha(\theta)$ dependent on the target model $\theta$. The rationale for this design is to construct an inference attack which exploits the similar statistics as in Attack S, in a more accurate way by computing it on the target model, yet with less computations (without the need to train shadow models). The hypothesis test is as follows.

$$\text{If } \ell(\theta, x_z, y_z) \leq c_\alpha(\theta), \text{ reject } H_0, \tag{17}$$

where the threshold function $c_\alpha(\theta)$ satisfies equation 10 and ensures false positive rate $\alpha$. Using equation 23, we rewrite the false positive rate in equation 10 as follows.

$$\mathbf{E}_{P_{H_0}(\theta)} \left[ \mathbf{E}_{\pi(z)} \left[ \mathbf{1}_{\ell(\theta, x_z, y_z) \leq c_\alpha(\theta)} \right] \right] = \alpha \tag{18}$$

One sufficient condition for equation 18 to hold is that, for any fixed target model $\theta$, we have

$$\mathbf{E}_{\pi(z)} \left[ \mathbf{1}_{\ell(\theta, x_z, y_z) \leq c_\alpha(\theta)} \right] = \alpha \tag{19}$$

To approximate the data distribution $\pi(z)$ in equation 19, the attacker samples the following set $P$ of population data points from distribution $\pi(z)$.

$$P = \{z_i\}_{i=1,2,...}, \text{ where } z_1, z_2, \cdots i.i.d. \sim \pi(z) \tag{20}$$

Therefore, we approximate $\pi(z)$ in equation 19 with the empirical distribution over samples $z \in P$.

$$\frac{|\{z \in P : \ell(\theta, x_z, y_z) \leq c_\alpha(\theta)\}|}{|P|} = \alpha \tag{21}$$

By solving equation 21 for every fixed target model $\theta$, we compute that $c_\alpha(\theta)$ equals the the $\alpha - $ percentile for the histogram of loss values $\ell(\theta, x_z, y_z)$ over population data samples $z = (x_z, y_z) \in P$. This recovers the model-dependent attack threshold $c_\alpha(\theta)$ for Attack P.

### 3.3 ATTACK R: SAMPLE-DEPENDENT MIA VIA REFERENCE MODELS

The privacy loss of the model with respect to the target data could be directly related to how susceptible the target data is to be memorized (e.g., being an outlier) Feldman (2020). Based on this finding, we design the membership inference Attack R that uses attack threshold function $c_\alpha(x_z, y_z)$ which depends on the target data (both its input features $x_z$ and the label $y_z$). This attack is very similar to the membership inference attacks designed for summary statistics, graphical models and machine learning, which use reference models to compute the probability of the null hypothesis Sankararaman et al. (2009); Murakonda et al. (2021); Long et al. (2020). The hypothesis test is as follows.

$$\text{If } \ell(\theta, x_z, y_z) \leq c_\alpha(x_z, y_z), \text{ reject } H_0, \tag{22}$$

where the threshold function $c_\alpha(x_z, y_z)$ satisfies equation 10 and ensures false positive rate $\alpha$. Using the chain rule of joint distribution, and the independence of $\theta$ and $z = (x_z, y_z)$ under null hypothesis $H_0$ in equation 1, we prove that

$$P_{H_0}(\theta, x_z, y_z) = P_{H_0}(x_z, y_z) \cdot P_{H_0}(\theta|x_z, y_z) = \pi(z) \cdot P_{H_0}(\theta), \tag{23}$$

where $P_{H_0}(\theta) = \mathbf{E}_{D \sim \pi^n}[P(\theta|D)]$ is the target model distribution specified by the null hypothesis $H_0$. Using equation 23, we rewrite the false positive rate constraint in equation 10 as follows.

$$\mathbf{E}_{\pi(z)}\left[\mathbf{E}_{P_{H_0}(\theta)}\left[\mathbf{1}_{\ell(\theta,x_z,y_z) \leq c_\alpha(x_z,y_z)}\right]\right] = \alpha \tag{24}$$

One sufficient condition for equation 24 to hold is that, for any fixed data point $z = (x_z, y_z)$,

$$\mathbf{E}_{P_{H_0}(\theta)}\left[\mathbf{1}_{\ell(\theta,x_z,y_z) \leq c_\alpha(x_z,y_z)}\right] = \alpha \tag{25}$$

To approximate the target model distribution $P_{H_0}(\theta)$ in equation 25, the attacker then samples the following set $R$ of reference models.

$$R = \{\theta_i\}_{i=1,2,\cdots}, \text{ where } \theta_i \sim \mathcal{T}(D_i), D_i \xleftarrow{n \; i.i.d.samples \sim \pi(z)} D_{pop} \tag{26}$$

Because the reference models $\theta_r = \{\theta_i\}_{i=1,2,\cdots}$ are trained on population datasets, they serve as samples from the distribution $P_{H_0}(\theta) = \mathbf{E}_{D \sim \pi^n}[P(\theta|D)]$. Therefore, we replace $P_{H_0}(\theta, z)$ in equation 25 with the empirical distribution over samples $\theta_i \in R$, as follows.

$$\frac{|\{\theta_i \in R : \ell(\theta_i, x_z, y_z) \leq c_\alpha(x_z, y_z)\}|}{|R|} = \alpha \tag{27}$$

By solving equation 27 for fixed target sample $z = (x_z, y_z)$, we compute that $c_\alpha(x_z, y_z)$ equals the the $\alpha$-percentile for the histogram of loss values $\ell(\theta_i, x_z, y_z)$ over reference models samples $\theta_i \in R$. This computes the sample-dependent attack threshold $c_\alpha(x_z, y_z)$ that in Attack R.

## 3.4 ATTACK D: MODEL-DEPENDENT AND SAMPLE DEPENDENT MIA VIA DISTILLATION

Can we design an attack that takes advantage of all the information available in the target model and the target data that can increase the chance of identifying the right hypothesis? We design a membership inference Attack D whose threshold function $c_\alpha(\theta, x_z, y_z)$ depends on both the target sample $z$ and the target model $\theta$, as follows.

$$\text{If } \ell(\theta, x_z, y_z) \leq c_\alpha(\theta, x_z, y_z), \text{ reject } H_0, \tag{28}$$

where the threshold function $c_\alpha(\theta, x_z, y_z)$ satisfies equation 10 and ensures false positive rate $\alpha$. However, the degree of freedom in the threshold function $c_\alpha(\theta, x_z, y_z)$ is still too large for us to directly solve equation 10. Therefore, we restrict $c_\alpha(\theta, x_z, y_z)$ to take the following form.

$$c_\alpha(\theta, x_z, y_z) = c_\alpha(D_\theta, x_z, y_z), \tag{29}$$

where $D_\theta = \mathcal{T}^{-1}(\theta)$ is the training dataset for target model $\theta$. For the simplicity of derivation, let us first assume that the randomized training algorithm $\mathcal{T}$ has a deterministic inverse mapping $\mathcal{T}^{-1} : \theta \to D$, i.e. the training dataset for a given model $\theta$ is uniquely specified. (Later we also show how to approximate the training datasets $D_\theta$ for model $\theta$ when the training algorithm $\mathcal{T}$ is not invertible.) Plugging equation 29 into equation 10, we rewrite the FPR constraint as follows.

$$\mathbf{E}_{P_{H_0}(\theta,x_z,y_z)}\left[\mathbf{1}_{\ell(\theta,x_z,y_z) \leq c_\alpha(D_\theta,x_z,y_z)}\right] = \alpha \tag{30}$$

By deterministic mapping from $\theta$ to $D_\theta$, the chain rule of joint distribution, and the independence between random variables $\theta$ and $z = (x_z, y_z)$ under null hypothesis $H_0$ in equation 1, we prove that

$$P_{H_0}(\theta, x_z, y_z) = P_{H_0}(\theta, x_z, y_z) \cdot 1 = P_{H_0}(\theta, x_z, y_z) \cdot P(D_\theta|\theta) \tag{31}$$

$$= P_{H_0}(D_\theta, \theta, x_z, y_z) = P_{H_0}(D_\theta, x_z, y_z) \cdot P_{H_0}(\theta|D_\theta, x_z, y_z) \tag{32}$$

$$= P_{H_0}(D_\theta, x_z, y_z) \cdot P(\theta|D_\theta) \tag{33}$$

where $P(\theta|D_\theta)$ is distribution of trained model under a fixed training dataset $D_\theta$, as specified in equation 5. Plugging equation 31 into equation 30, we rewrite the false positive rate constraint as

$$\mathbf{E}_{P_{H_0}(D_\theta,x_z,y_z)}\left[\mathbf{E}_{P(\theta|D_\theta)}\left[\mathbf{1}_{\ell(\theta,x_z,y_z) \leq c_\alpha(D_\theta,x_z,y_z)}\right]\right] = \alpha \tag{34}$$

One sufficient condition for equation 34 to hold is that, for any fixed sample $z$ and target model $\theta$,

$$\mathbf{E}_{P(\theta'|D_\theta)}\left[\mathbf{1}_{\ell(\theta',x_z,y_z) \leq c_\alpha(D_\theta,x_z,y_z)}\right] = \alpha, \tag{35}$$

where $D_\theta$ is an implicitly fixed training dataset for the target model $\theta$, and the distribution $P(\theta'|D_\theta)$ captures retrained models on the training dataset $D_\theta = \mathcal{T}^{-1}(\theta)$ for given target model $\theta$, as follows.

$$\mathcal{T}\left(\mathcal{T}^{-1}(\theta)\right) = \theta' \text{ with probability } P(\theta'|D_\theta). \tag{36}$$

To approximate this distribution of retrained model $\mathcal{T}\left(\mathcal{T}^{-1}(\theta)\right) \sim P(\theta'|D_\theta)$, the attacker generates the following set of self-distilled models using the target model $\theta$.

$$M = \{\theta_i\}_{i=1,2,\cdots}, \text{ where } \theta_i \sim \mathcal{T}(D_i), D_i \xleftarrow{\text{soft-labeled with } \theta} D_i' \xleftarrow{i.i.d.samples \sim \pi(z)} D_{pop} \tag{37}$$

These distilled models $M = \{\theta_i\}_{i=1,2,\cdots}$ approximate samples from the retrained model distribution $\mathcal{T}(\mathcal{T}^{-1}(\theta)) \sim P(\theta|D_\theta)$. This is because $\theta_i$ is trained on distillation dataset $D_i$ consisting of population data points which are soft-labeled with the target model $\theta$. This roughly recovers the target model $\theta$ trained on $D_\theta$, however without its potential dependence on $z$. Therefore, the attacker approximate $P(\theta|D_\theta)$ in the false positive rate constraint equation 35 with the empirical distribution over distilled models samples in $M$, as follows.

$$\frac{|\{\theta_i \in M : \ell(\theta_i, x_z, y_z) \le c_\alpha(D_\theta, x_z, y_z)\}|}{|M|} = \alpha \tag{38}$$

By solving equation 38 for fixed target model $\theta$ and target point $z$, we compute that $c_\alpha(D_\theta, x_z, y_z)$ equals the the $\alpha$-percentile for the histogram of loss values $\ell(\theta_i, x_z, y_z)$ on distilled models $\theta_i \in M$. This recovers the model-dependent and sample-dependent attack threshold $c_\alpha(x_z, y_z)$ in Attack D.

### 3.5 ATTACK L: LEAVE ONE OUT ATTACK

An ideal attack, that removes the randomness over the training data (except the target data that could potentially be part of the training set) would be the *leave one out* attack. In this attack, the adversary trains reference models $\theta'$ on $D \setminus \{(x_b, y_b)\}$. The attack would be in the same class of attacks as in Attack D, as it will be a model-dependent and data-dependent attack. It also runs a similar hypothesis test, however the attack requires assuming the adversary already knows the $n-1$ data records in $D \setminus \{(x_b, y_b)\}$. This is a totally acceptable assumption in the setting of privacy auditing.

Note that Attack D aims at reproducing the results of the leave-one-out attack without assuming the knowledge of $n-1$ data records in $D \setminus \{(x_b, y_b)\}$.

### 3.6 SUMMARY AND COMPARISON OF DIFFERENT ATTACKS

For identifying whether a data point $z$ has been part of the training set of $\theta$, here are the main underlying questions for the attacks we present in this section:

• How likely is the loss $\ell(\theta, z)$ to be a sample from the distribution of loss of random samples from the population on (Attack P: the same model) (Attack S: models trained on the population data)? Depending on the tolerable false positive rate $\alpha$ and the estimated distribution of loss, we reject the null hypothesis.

• How likely is the loss $\ell(\theta, z)$ to be a sample from the distribution of loss of $z$ on (Attack R: models trained on population data) (Attack D: models trained to be as close as possible to the target model, using distillation) (Attack L: models trained on $n-1$ records from $D$ excluding $z$)? Depending on the tolerable false positive rate $\alpha$ and the estimated distribution of loss, we reject the null hypothesis.

Effectively, these questions cover different types of hypothesis tests that could be designed for performing membership inference attacks. We expect these attacks to have a different error due to the uncertainties that can influence their performance.

Attacks S and P are of the same nature. However, attack S could potentially have a higher error due to its imprecision in using other models to approximate the loss distribution of the target model on population data. Attacks R, D, and L are also of the same nature. However, we expect attacks D to have more confidence in the tests due to reducing the uncertainty of other training data that can influence the model's loss. Thus, we expect attack D to be the closest to the strongest attack which is the leave-one-out attack.

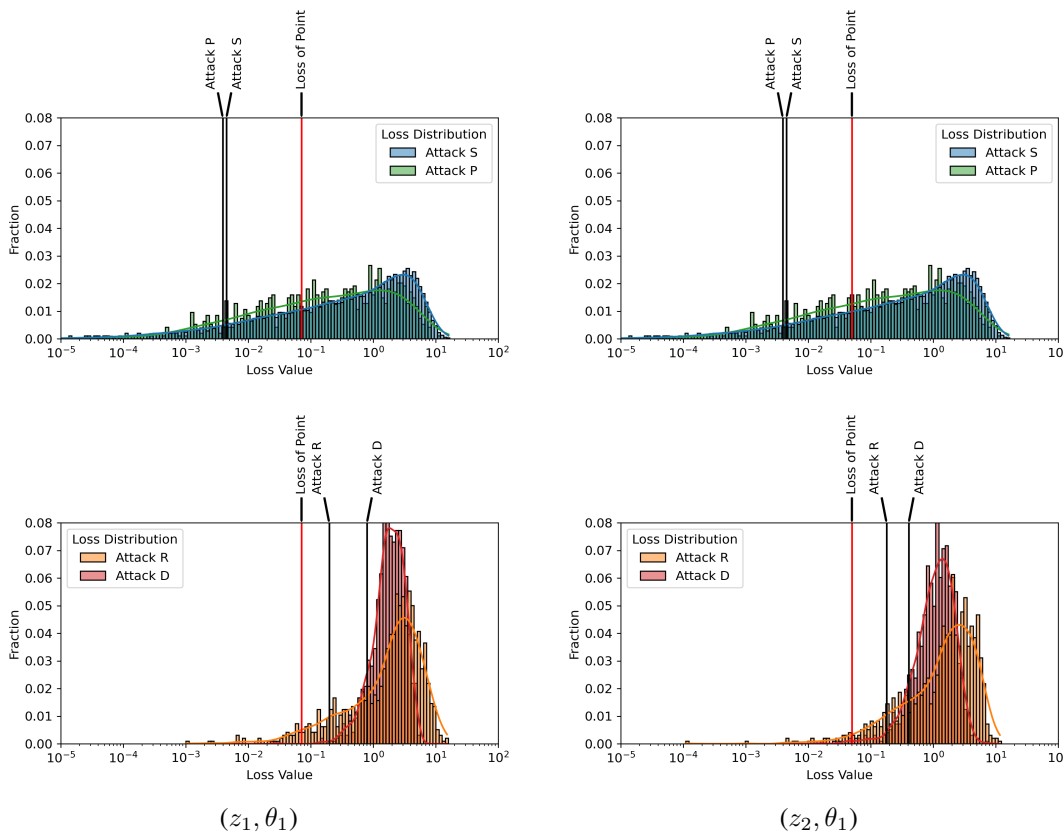

$$(z_1, \theta_1) \qquad\qquad (z_2, \theta_1)$$

Figure 1: Loss distributions used by Attack S, P, R, D for points $z_1, z_2$ and target model $\theta_1$ in Purchase100 Configuration IIa, $\alpha = 0.1$. Note that both $z_1$ and $z_2$ are datapoints from the same class. We also show the loss distributions for $z_1, z_2$ on another target model $\theta_2$ in the appendix.

## 4 EXPERIMENTS

In this section, we empirically measure and study the performance of different attack strategies that we derived from our framework. A detailed analysis of the internal mechanics of these attacks and how they differ from each other is provided in the Appendix A.2.

**Evaluation of attacks performance** Which one of the attacks S, P, R, D has the best performance? How to evaluate the strength of attack besides attack accuracy? We quantify the attacker's performance using two metrics: its true positive rate (TPR), and its false positive rate (FPR), over the random samples from member and non-member data and random target models. The ROC curve captures the tradeoff between the TPR and FPR of an attack, as its threshold $c_\alpha$ is varied across different FPR tolerance $\alpha$. The AUC (area under the ROC curve) score measures the strength of an attack. Therefore, we plot the ROC curves of all attacks on the Purchase100 dataset, and computes their AUC (area under the ROC curve) score in Figure 3 (in the appendix). The attack with the highest AUC score on Purchase100 is Attack D, which has the least level of uncertainty, as discussed Section 3. Table 1 and Table 4 shows more detailed AUC score results for different attacks under more settings.

**Comparison between the Attacks** Besides attack strength, how differently are the attacks performing on random input target models and target points? How do the different internal mechanics of attacks changing the attacker's guess qualitatively? Do the attacks succeed on similar or different samples of member data and target model? Do the attacks have different level of confidence on the same input? How far away are the attack performance from the most ideal leave-one-out attacks described in Section 3? Answers to these questions require understanding how and why the attacks perform differently, for which we do detailed comparisons between attacks as follows.

|      | Train Acc.       | Test Acc.        | Attack S          | Attack P          | Attack R          | Attack D           |
|------|------------------|------------------|-------------------|-------------------|-------------------|--------------------|
| Ia   | $96.2 \pm 0.031$ | $52.5 \pm 0.026$ | $\mathbf{0.809} \pm 0.017$ | $\mathbf{0.822} \pm 0.014$ | $\mathbf{0.84} \pm 0.023$ | $\mathbf{0.876} \pm 0.009$ |
| Ib   | $51.3 \pm 0.153$ | $35.0 \pm 0.095$ | $\mathbf{0.628} \pm 0.035$ | $\mathbf{0.646} \pm 0.026$ | $\mathbf{0.643} \pm 0.045$ | $\mathbf{0.652} \pm 0.045$ |
| IIa  | $99.5 \pm 0.004$ | $65.4 \pm 0.009$ | $\mathbf{0.752} \pm 0.008$ | $\mathbf{0.755} \pm 0.006$ | $\mathbf{0.799} \pm 0.009$ | $\mathbf{0.821} \pm 0.004$ |
| IIb  | $64.1 \pm 0.039$ | $49.9 \pm 0.019$ | $\mathbf{0.599} \pm 0.009$ | $\mathbf{0.609} \pm 0.009$ | $\mathbf{0.613} \pm 0.012$ | $\mathbf{0.635} \pm 0.004$ |
| III  | $100.0 \pm 0.0$  | $75.5 \pm 0.004$ | $\mathbf{0.687} \pm 0.003$ | $\mathbf{0.687} \pm 0.003$ | $\mathbf{0.755} \pm 0.004$ | $\mathbf{0.768} \pm 0.002$ |
| IV   | $95.74 \pm 0.01$ | $71.71 \pm 0.009$| $\mathbf{0.647} \pm 0.004$ | $\mathbf{0.55} \pm 0.005$ | $\mathbf{0.682} \pm 0.009$ | $\mathbf{0.70} \pm 0.005$ |

Table 1: AUC Scores of all attacks on Purchase100 Dataset. Configurations Ia, Ib are trained on 2500 datapoints, configurations IIa, IIb are trained on 5000 datapoints, and configurations III, IV are trained on 10000 datapoints. Configurations Ib, IIb are trained using L2 regularization with regularization penalty $\lambda = 0.01$. Configuration IV is trained with a gradient clipping norm of 2.0. Configurations Ia, Ib, IIa, IIb, and III use $n = 1000$ (shadow, reference, distilled) models, whereas configuration IV uses $n = 30$ (shadow, reference, distilled) models.

• **Similarity of attacks with each other.** The scatter plot comparing Attack S and P in Figure 5 is roughly linearly centered around the diagonal, with slightly more points in NorthWest then in SouthEast. This shows that Attack S and Attack P almost always guesses membership of train points similarly, while Attack P performs slightly better than Attack S. Meanwhile, Attack D dominates Attack R for correctly guessing membership of train points because their comparison scatter plot is biased towards the NorthWest direction.

• **Gap between attacks to the ground truth.** From Table 2, among all attacks, Attack D agrees with the ground truth the most on train points, while the least on test points. This matches our observation that Attack D has a larger threshold under given $\alpha$ from Figure 1, which causes it to guess both more points as members. Besides that, we see that the agreement rate between Attack S and Attack P is as high as $0.94$ both on train points and test points, this matches their linear comparison scatter plot in Figure 5. This may be because Attack P and Attack R both reduces one degree of uncertainty for the joint distribution under null hypothesis $H_0$, as discussed in Section 3.

• **Closeness of attacks to ideal leave-one-out attack.** One interesting observation from Table 2 is that, among all the attacks, Attack D agrees with the Attack L the most, with agreement rate $0.98$ on train points and $0.82$ on test points. We believe this is because Attack D is highly similar in nature with Attack L, by approximating the training dataset of a target model and performing retraining, as discussed in Section 3.

|     | GT    | L     | S     | P     | R     | D     |
|-----|-------|-------|-------|-------|-------|-------|
| GT  |       | 0.968 | 0.696 | 0.662 | 0.772 | 0.992 |
| L   | 0.41  |       | 0.692 | 0.654 | 0.796 | 0.968 |
| S   | 0.656 | 0.73  |       | 0.874 | 0.592 | 0.696 |
| P   | 0.664 | 0.706 | 0.948 |       | 0.586 | 0.662 |
| R   | 0.656 | 0.722 | 0.756 | 0.732 |       | 0.78  |
| D   | 0.372 | 0.822 | 0.716 | 0.708 | 0.692 |       |

Table 2: Agreement rate between ground truth (GT) membership values, and Attacks L, S, P, R, D for 500 train and 500 test datapoints. The upper triangle of the table corresponds to the agreement rates of train datapoints, whereas the lower triangle corresponds to the agreement rates of test datapoints. The experimental setup is Purchase100 Configuration IIa, with $\alpha = 0.3$.

## 5 SUMMARY

We provide a framework for auditing the privacy risk from machine learning models through membership inference attacks. The framework is used to derive attack strategies and also highlight the factors beyond leakage from the models that affect the attack performance. We also empirically analyze the performance of these attack strategies against models trained on benchmark datasets.

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

## A    APPENDIX

### A.1    DETAILED DERIVATION OF APPROXIMATED LRT FOR MEMBERSHIP INFERENCE

We first prove two useful approximation inequalities about the posterior distribution $P(\theta|D)$ as follows.

1. For arbitrary data point $z = (z_x, z_y)$, and arbitrary dataset $D$, we have

$$P(\theta|D \cup z) \geq e^{-\frac{\ell(\theta, z_x, z_y)}{T}} \cdot P(\theta|D). \qquad (39)$$

2. Let $z_1, z_2, \cdots, z_n$ be i.i.d. samples from the data distribution $\pi(z)$. Then when $n$ is large enough, for any model parameter $\theta$, we have

$$\mathbf{E}_{z_1,\cdots,z_n \sim \pi}[P(\theta|z_1,\cdots,z_n)] \approx \mathbf{E}_{z_1,\cdots,z_{n-1} \sim \pi}[P(\theta|z_1,\cdots,z_{n-1})] \tag{40}$$

We now offer details for deriving the approximated likelihood ratio test (LRT) for membership inference.

**Definition A.1 (Approximated LRT for membership inference)** *Let $(\theta, z)$ be random samples from the joint distribution of target model and target data point, specified by one of the following membership hypotheses.*

$$H_0 : D \xleftarrow{n \ i.i.d. samples \sim \pi(z)} D_{pop}, \theta \sim \mathcal{T}(D), z \xleftarrow{sample \sim \pi(z)} D_{pop} \tag{41}$$

$$H_1 : D \xleftarrow{n \ i.i.d. samples \sim \pi(z)} D_{pop}, \theta \sim \mathcal{T}(D), z \xleftarrow{sample} D \tag{42}$$

*The likelihoods function of hypothesis $H_0$ and $H_1$ given observed target model $\theta$ and target data point $z$ is as follows.*

$$L(H_0|\theta, z) = P_{H_0}(\theta, z) = \pi(z) \cdot \mathbf{E}_{D \sim \pi^n}[P(\theta|D)] \tag{43}$$

$$L(H_1|\theta, z) = P_{H_1}(\theta, z) = \sum_D P_{H_1}(D, \theta, z) = \sum_D \pi(z) \cdot P_{H_1}(D|z) \cdot P(\theta|D) \tag{44}$$

$$= \pi(z) \cdot \mathbf{E}_{D' \sim \pi^{n-1}}[P(\theta|D' \cup z)] \tag{45}$$

*(By equation 39)* $\geq \pi(z) \cdot e^{-\frac{1}{T}\ell(\theta, z_x, z_y)} \cdot \mathbf{E}_{D' \sim \pi^{n-1}}[P(\theta|D')] \tag{46}$

*(By equation 40)* $\approx \pi(z) \cdot e^{-\frac{1}{T}\ell(\theta, z_x, z_y)} \cdot \mathbf{E}_{D \sim \pi^n}[P(\theta|D)] \tag{47}$

*Therefore the LRT statistics is*

$$\lambda(\theta, z) = \frac{L(H_0|\theta, z)}{L(H_1|\theta, z)} \leq e^{\frac{1}{T}\ell(\theta, z_x, z_y)} \frac{\mathbf{E}_{D \sim \pi^n}[P(\theta|D)]}{\mathbf{E}_{D' \sim \pi^{n-1}}[P(\theta|D')]} \approx e^{\frac{1}{T}\ell(\theta, z_x, z_y)} \tag{48}$$

*The LRT hypothesis test rejects $H_0$ when the LRT statistic is small. By equation 6, the rejection region $\{(\theta, z) : \lambda(\theta, z) \leq c\}$ can be approximated as follows.*

$$\left\{ (\theta, z) : \ell(\theta, z_x, z_y) \leq T \cdot \log c \right\} \tag{49}$$

**Datasets**

- The **Purchase100** dataset is based on Kaggle's "Acquire Valued Shoppers Challenge" that contains shopping histories for thousands of individuals[4]. We use a simplified preprocessed purchase dataset provided in Shokri et al. (2017). There are $197,324$ data points in this dataset, where each data point has $600$ binary features. The data points are clustered into $100$ classes to represent different shopping style.

- The **CIFAR10** and **CIFAR100** datasets are widely used benchmark datasets for image classification. The CIFAR10 dataset consists of $60,000$ data points in 10 classes, whereas the CIFAR100 dataset consists of $60,000$ data points in 100 classes. Here each data point is a $32 \times 32$ color image, and there are $50,000$ training images and $10,000$ test images in total.

- The **MNIST** dataset is a widely used benchmark dataset for image classification. The MNIST dataset consists of $70,000$ data points in 10 classes, where each data point is a $28 \times 28$ handwritten digit image. In total, there are $60,000$ training images and $10,000$ test images.

**Experiment setup**

1. For each configuration, we train up to $n = 1000$ models on random IID splits of the data. For the Purchase100 configurations, we use a 4 layer MLP with layer units = [512, 256,

---

[4]https://www.kaggle.com/c/acquire-valued-shoppers-challenge

128, 64]. For CIFAR100 and MNIST configurations, we use a 2 layer CNN with filters = [32, 64] and max pooling. For CIFAR10 configurations, we use AlexNet and a 3-block VGGNet. For all target models and datasets, we use SGD as the optimization function and categorical crossentropy loss. For Configuration IV of Purchase100, we use gradient clipping while training the models. For CIFAR10, we add momentum while training the models.

2. Target models: We use the first 10 models as target models on each of the Purchase100, CIFAR10, CIFAR100 and MNIST datasets. The performance of Attacks S, P, and R are evaluated on all 10 models, whereas Attack D is evaluated on the first 3 models.

3. Shadow models for Attack S and Reference models for Attack R: Given a target model, we use the remaining $(n-1) = 999$ models as shadow models or reference models for Attack S and Attack R respectively.

4. Distilled models for Attack D: Given a target model, we use the same random IID splits used to train $n = 1000$ models to distill $(n-1) = 999$ models using the soft labels from the target model.

5. Using $n < 1000$ models: For Configuration IV of Purchase100 and Configuration III of CIFAR10, we report the attack performance results using $n = 30$ shadow, reference, and distilled models. For Configurations I and II of CIFAR10 we report the attack performance results using $n = 400$ shadow, reference, and distilled models.

## A.2 Additional Empirical Evaluation Results

In this section, we provide a detailed analysis of the internal workings of the attack strategies. We specifically compare how the thresholds on loss values derived from these strategies differ on various target points and target models.

**Illustration of Attack threshold** How differently does the attack threshold in Attack S, P, R, D depend on the target model $\theta$ and the target data point $z$? How is the different dependence of threshold on $\theta$ and $z$ in different attacks affecting their success? To investigate these important questions, we plot the loss histograms that different attacks use to compute the thresholds, on different input $\theta_1, \theta_2$ and $z_1, z_2$ in Figure 1. The loss histogram approximates the LRT statistics distribution under null hypothesis $H_0$ in equation 1, with different uncertainty for different attacks, as discussed in 3. We see that model-dependence and sample-dependence of attack threshold reduces the uncertainty in the loss histogram. The attacks (Attack D) with more concentrated loss histogram are more likely to succeed.

|     | Train Acc.      | Test Acc.       | Attack S          | Attack P          | Attack R          | Attack D          |
| --- | --------------- | --------------- | ----------------- | ----------------- | ----------------- | ----------------- |
| I   | $96.2 \pm 0.046$ | $40.9 \pm 0.029$ | $0.870 \pm 0.018$ | $0.857 \pm 0.023$ | $0.874 \pm 0.018$ | $0.871 \pm 0.007$ |
| II  | $97.8 \pm 0.012$ | $45.9 \pm 0.010$ | $0.860 \pm 0.014$ | $0.868 \pm 0.008$ | $0.858 \pm 0.019$ | $0.889 \pm 0.011$ |
| III | $97.4 \pm 0.004$ | $68.2 \pm 0.011$ | $0.706 \pm 0.011$ | $0.708 \pm 0.009$ | $0.737 \pm 0.014$ | $0.742 \pm 0.003$ |

Table 3: AUC Scores of all attacks on CIFAR10 Dataset. Configuration I is trained on 2500 datapoints, and configuration II is trained on 5000 datapoints. Configurations I and II are trained using AlexNet. Configuration III is trained on 10000 datapoints using a 3-block VGGNet. Here we report the results of the attacks for Configurations I and II using $n = 400$ (shadow, distilled, reference) models, and $n = 30$ for Configuration III. All models have been trained using the SGD optimizer with momentum, and an L2 regularization penalty $\lambda = 0.001$.

## B Related Work

The current work in this domain can be broadly grouped into three categories: 1) Empirical works for improving the existing attack strategies or adapting them to different settings and models 2) Theoretically/empirically analyzing the privacy risk in various systems using the existing attack strategies 3) Exploring the connections with differential privacy and using them to establish lower bounds on leakage and/or select privacy parameters. Below, we provide a brief summary of works in all the three categories.

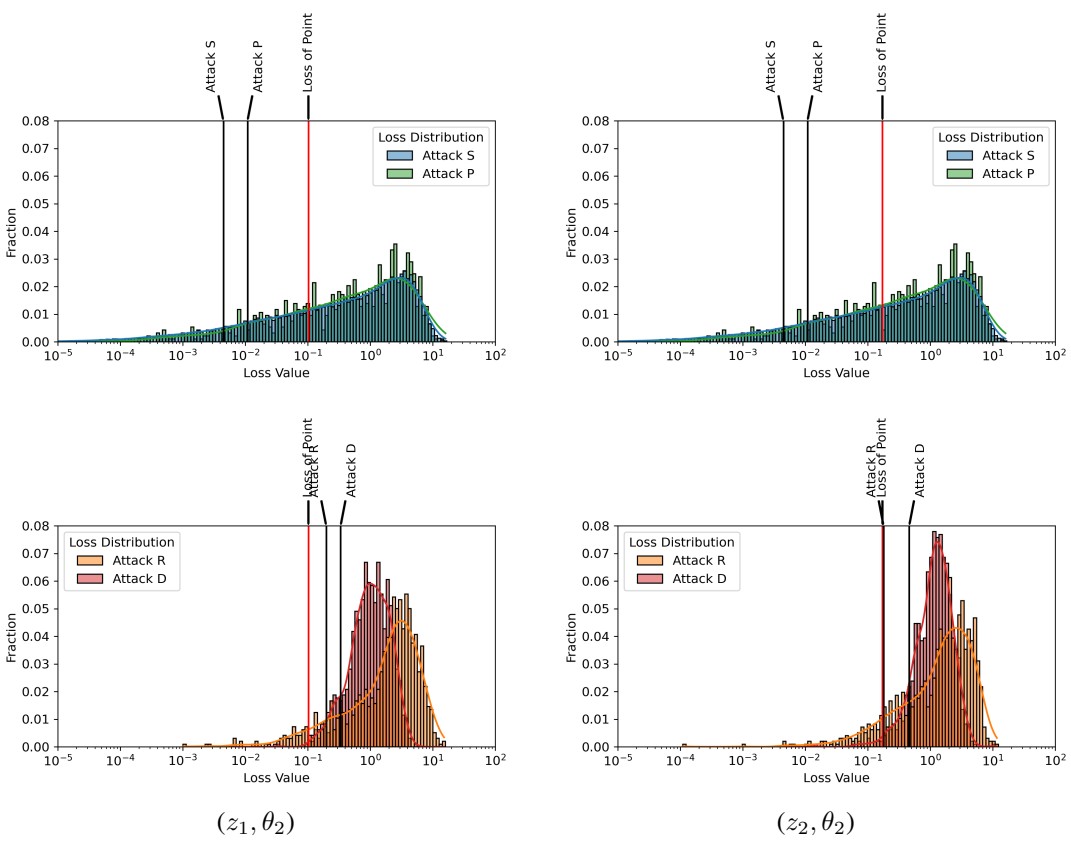

Figure 2: Loss distributions used by Attack S, P, R, D for points $z_1, z_2$ and target models $\theta_1, \theta_2$ in Purchase100 Configuration IIa, $\alpha = 0.1$. Note that both $z_1$ and $z_2$ are datapoints from the same class and are members of both the target models' training datasets. We show the loss distributions for $z_1, z_2$ on target model $\theta_1$ in the main paper, and on $\theta_2$ here.

|      | Train Acc.     | Test Acc.       | Attack S          | Attack P          | Attack R          | Attack D             |
|------|----------------|-----------------|-------------------|-------------------|-------------------|----------------------|
| Ia   | $97.4 \pm 0.013$ | $14.9 \pm 0.009$  | $\mathbf{0.959} \pm 0.005$ | $\mathbf{0.960} \pm 0.004$ | $\mathbf{0.964} \pm 0.006$ | $\mathbf{0.957} \pm 0.0003$ |
| Ib   | $89.8 \pm 0.035$ | $13.9 \pm 0.011$  | $\mathbf{0.937} \pm 0.007$ | $\mathbf{0.941} \pm 0.004$ | $\mathbf{0.938} \pm 0.012$ | $\mathbf{0.941} \pm 0.004$  |
| IIa  | $97.9 \pm 0.006$ | $20.4 \pm 0.006$  | $\mathbf{0.944} \pm 0.004$ | $\mathbf{0.945} \pm 0.003$ | $\mathbf{0.945} \pm 0.006$ | $\mathbf{0.936} \pm 0.0$   |
| IIb  | $87.9 \pm 0.020$ | $19.63 \pm 0.007$ | $\mathbf{0.905} \pm 0.007$ | $\mathbf{0.906} \pm 0.007$ | $\mathbf{0.904} \pm 0.008$ | $\mathbf{0.893} \pm 0.002$  |

Table 4: AUC Scores of all attacks on CIFAR100 Dataset. Configurations Ia, Ib are trained on 2500 datapoints, and configurations IIa, IIb are trained on 5000 datapoints. Configurations Ib, IIb are trained using L2 regularization with regularization penalty $\lambda = 0.005$.

|      | Train Acc.     | Test Acc.       | Attack S          | Attack P          | Attack R          | Attack D          |
|------|----------------|-----------------|-------------------|-------------------|-------------------|-------------------|
| Ia   | $97.9 \pm 0.004$ | $95.8 \pm 0.007$  | $\mathbf{0.50} \pm 0.004$  | $\mathbf{0.50} \pm 0.005$  | $\mathbf{0.557} \pm 0.009$ | $\mathbf{0.549} \pm 0.006$ |
| Ib   | $94.2 \pm 0.005$ | $93.4 \pm 0.006$  | $\mathbf{0.497} \pm 0.005$ | $\mathbf{0.496} \pm 0.004$ | $\mathbf{0.544} \pm 0.014$ | $\mathbf{0.522} \pm 0.005$ |
| IIa  | $98.6 \pm 0.001$ | $97.1 \pm 0.002$  | $\mathbf{0.496} \pm 0.005$ | $\mathbf{0.496} \pm 0.006$ | $\mathbf{0.551} \pm 0.011$ | $\mathbf{0.544} \pm 0.004$ |
| IIb  | $94.6 \pm 0.003$ | $94.5 \pm 0.003$  | $\mathbf{0.491} \pm 0.005$ | $\mathbf{0.49} \pm 0.005$  | $\mathbf{0.52} \pm 0.011$  | $\mathbf{0.497} \pm 0.004$ |

Table 5: AUC Scores of all attacks on MNIST Dataset. Configurations Ia, Ib are trained on 2500 datapoints, and configurations IIa, IIb are trained on 5000 datapoints. Configurations Ib, IIb are trained using L2 regularization with regularization penalty $\lambda = 0.05$.

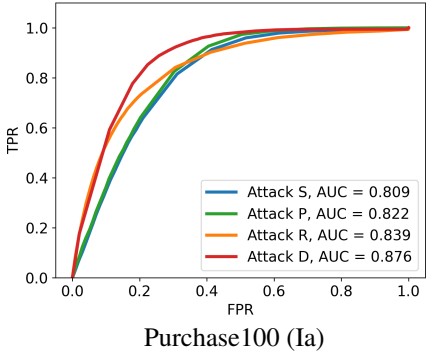 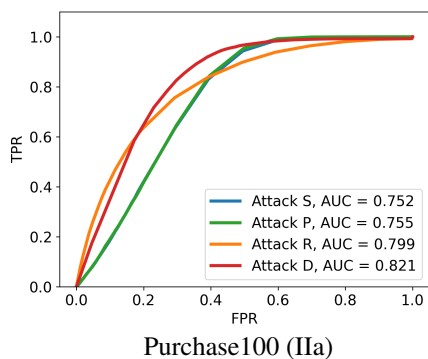

Purchase100 (Ia)  Purchase100 (IIa)

Figure 3: FPR vs TPR with AUC scores for all attacks on Configurations Ia and IIa of the Purchase100 dataset. Note that models in Configuration Ia have been trained using 2500 points, and models in Configuration IIa have been trained using 5000 points.

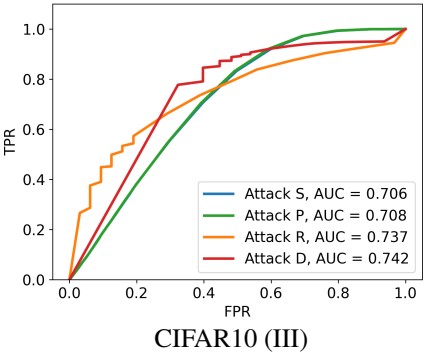 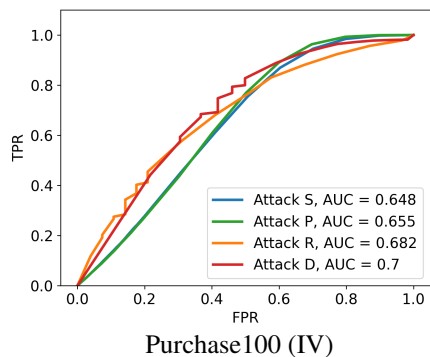

CIFAR10 (III)  Purchase100 (IV)

Figure 4: FPR vs TPR with AUC scores for all attacks on Configuration III of the CIFAR10 dataset and Configuration IV of the Purchase100 dataset. Note that models in Configuration III of CIFAR10 use the 3-block VGGNet architecture. Models in Configuration IV of Purchase100 have been trained using gradient clipping with an L2 clipping norm of 2.0. All attacks here use $n = 30$ (shadow, reference, distilled) models.

**Empirical attack strategies for membership inference:** Shokri et al. (2017) demonstrated the vulnerability of machine learning models to membership inference attacks in the black-box setting, where the adversary has only a query access to the target model. The attack algorithm is based on the concept of shadow models, which are models trained on some attacker dataset that is similar to that of the training data. Membership inference is modeled as a binary classification task for an attack model that is trained on the predictions of shadow models on the attacker dataset. A huge literature followed this work extending the attacks to different setting like white box access (Nasr et al., 2019; Sablayrolles et al., 2019; Leino & Fredrikson, 2020), label-only access (Li & Zhang, 2020; Choquette-Choo et al., 2021), federated learning (Melis et al., 2019), transfer learning (Zou et al., 2020) and different types of data, models such as aggregate location data (Pyrgelis et al., 2017), generative models (Hayes et al., 2019), language models (Carlini et al., 2019; Song & Shmatikov, 2019; Carlini et al., 2020), sentence embeddings (Song & Raghunathan, 2020), and speech recognition models (Shah et al., 2021). Multiple works have looked at improving the attack methodology through a more fine grained analysis or by reducing the background knowledge and the compute power required to execute the attack (Long et al., 2018; Song & Mittal, 2021; Salem et al., 2018). See (Hu et al., 2021) for a more comprehensive list of membership inference attacks against machine learning models. All these works follow the same attack framework for membership inference but they either exploit a slightly different signal that is correlated with membership of a point in the training set or find an efficient way to exploit the already known signals.

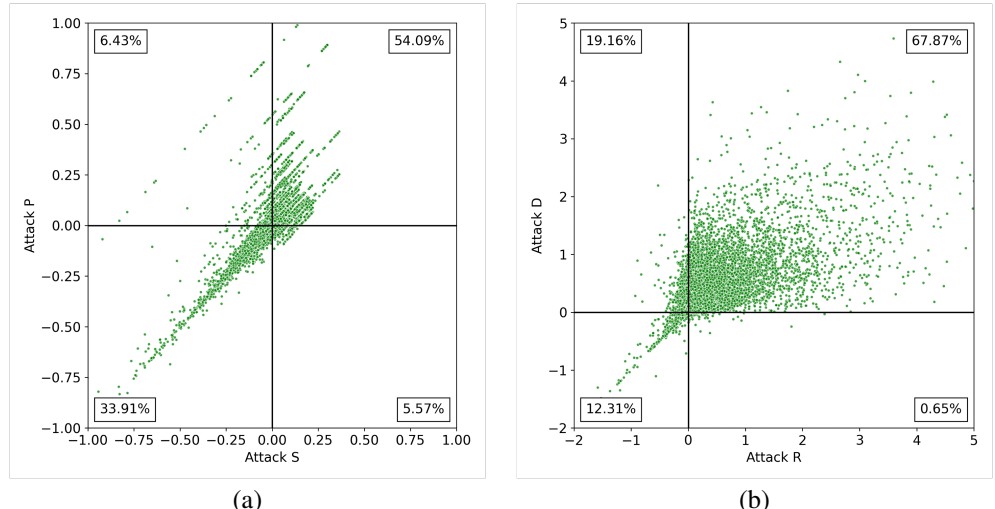

Figure 5: Scatterplots comparing Attacks S and P, and Attacks R and D in Purchase100 Configuration IIa. Each dot on a scatterplot corresponds to a particular train datapoint. The two coordinates of the dot are for the two attacks being compared. Each coordinate of the dot corresponds to the difference between the loss of the point on the target model and the loss threshold used by the particular attack on the point. Plot (a) compares Attacks S and P, whereas plot (b) compares Attacks R and D.

**Privacy risk analysis with membership inference attacks:** Homer et al. (2008) performed the first membership inference attack on genome data to identify the presence of an individual's genome in a mixture of genomes. Sankararaman et al. (2009) provided a formal analysis of this risk of detecting the presence of an individual from aggregate statistics computed on independent and binary attributes. Murakonda et al. (2021) extended this analysis to the case of releasing Discrete Bayesian networks learned from data with dependent attributes. Backes et al. (2016) perform an analysis similar to that of (Sankararaman et al., 2009) but for MicroRNA data (aggregate statistics computed on independent and continuous attributes). Dwork et al. (2015) provide a more extensive analysis when the released statistics are noisy and the attacker has only one reference sample to perform the attack. The key results of all these works establish the privacy risk of releasing aggregate statistics by quantifying the success of membership inference attacks as a function of the number of statistics released and the number of individuals in the dataset. Similar attempts were made to analyze the privacy risk of machine learning models through membership inference via the lens of mutual information (Farokhi & Kaafar, 2020) and generalization error (Yeom et al., 2018; Del Grosso et al., 2021). Beyond these theoretical analyses, membership inference attacks are also used to empirically study the trade-offs between privacy and other desirable characteristics for machine learning models such as fairness (Chang & Shokri, 2020), robustness to adversarial examples (Song et al., 2019), and providing explanations (Shokri et al., 2021).

**Differential privacy and Membership inference:** The definitions of differential privacy (Dwork et al., 2006) and membership inference (Homer et al., 2008; Dwork et al., 2015; Shokri et al., 2017) are very closely connected and the hypothesis testing interpretation of differential privacy provides a clear view of the relationship between them. Satisfying differential privacy is equivalent to imposing a bound on the ability to distinguish any two neighboring datasets that differ by the presence of one individual i.e., inferring about the presence/absence of the individual. The bound is stated as a trade-off between the type-I and type-II errors in distinguishing the two neighboring datasets. Hypothesis testing interpretation of differential privacy is very useful in deriving tight compositions (Kairouz et al., 2015) and has even motivated a new relaxed notion of differential privacy called f-DP (Dong et al., 2019). By definition, differentially private algorithms bound the success of membership inference attacks. Multiple works (Yeom et al., 2018; Erlingsson et al., 2019; Humphries et al., 2020), each improving on the previous work, have provided upper bounds on the success of membership inference attacks as a function of the parameters in differential privacy. Jayaraman & Evans (2019) evaluated the performance of membership inference attacks on machine learning models trained

with different values of epsilon under different relaxed notions of differential privacy. Rahman et al. (2018) also use membership inference attacks to measure the privacy loss on models trained with differentially private algorithms. The empirical performance of membership inference attacks has also been used to provide lower bounds on the privacy guarantees achieved by various differentially private algorithms (Jagielski et al., 2020; Nasr et al., 2021; Malek et al., 2021). The key difference between the empirical analysis of membership inference in the previous three works and other works is that they simulate the exact adversary in differential privacy i.e., they train multiple models with and without one particular training point and keep the rest of training set fixed. The performance of the attack (type I and type II errors) is averaged over these models, whereas in the previous works the model is fixed and the performance is computed as an average over points in the training and test sets. This simulation of the DP adversary helps in removing the effects of other points in the dataset when measuring the leakage through the model about a particular point of interest.

