# OpenReview forum: "Privacy Auditing of Machine Learning using Membership Inference Attacks"
_ICLR.cc/2022/Conference — ICLR 2022 Submitted_

### Official Review · Reviewer_BGkr · 2021-11-01

**Correctness:** 3
**Technical Novelty And Significance:** 3
**Empirical Novelty And Significance:** 3
**Recommendation:** 5
**Confidence:** 4

**Main Review:**

Strengths

The hypothesis testing framework is well-motivated and the formulation is solid which also covers prior work such as shadow model MIAs. The attacks based on distillation also seem noval and not discussed before. The authors also showed that the newly proposed attacks are better than shadow model MIAs.

Weaknesses and questions:
1. Many of the attacks could still be inefficient for auditing purposes as one might need to train 1000 reference models to perform the attacks. The numbers also are only high for models that are extremely overfitting (which is also true for previous work). How could we audit models that are well-generalized?
2. The comparison and difference between this work and prior work is vaguely discussed. No in detailed comparison with [1], which also formalized MIA with indistinguishability based framework. The hypothesis testing framework in this work is very similar to the threshold adversary in [1]. I hope authors could discuss the differences in detail. No comparison to white-box models (e.g. [2]). How would this compare to the proposed framework?
Minor: related works should be discussed in the main text instead of supplementary materials.
3. No connection made between this work and differential privacy, although the indistinguishability setup is highly related. It would be nice to see how the framework translates to privacy loss in DP.
4. Minor: column names overflowed in Table 4.

References

[1] Privacy Risk in Machine Learning: Analyzing the Connection to Overfitting. Samuel Yeom, Irene Giacomelli, Matt Fredrikson, Somesh Jha

[2]  Comprehensive Privacy Analysis of Deep Learning: Passive and Active White-box Inference Attacks against Centralized and Federated Learning. Milad Nasr, Reza Shokri, Amir Houmansadr


**Summary Of The Paper:**

This paper proposed a hypothesis testing framework for membership inference attacks (MIAs). The framework involves different algorithms for finding the loss threshold for MIAs, i.e. if a target data point with loss less than the threshold value then it will be inferred as a member of training data. The notation of membership privacy loss is the same as in differential privacy and the proposed algorithms improves the utility of the attacks compared to prior works on benchmark datasets.


**Summary Of The Review:**

This paper is well-motivated and the proposed framework could potentially be impactful. However, there are many empirical limitations on how to apply the attacks for auditing state of the art models. Furthermore, the evaluation and comparison of baselines methods are not thorough. I therefore recommend a weak reject.

---

> ### Author Response · Authors · 2021-11-17
> **Response to reviewer BGkr**
>
> **Q1**
> > Many of the attacks could still be inefficient for auditing purposes as one might need to train 1000 reference models to perform the attacks. The numbers also are only high for models that are extremely overfitting (which is also true for previous work). How could we audit models that are well-generalized?
>
> Thanks for the comment.
> - We add the following attack performance results for well-generalized models:
>   - (a) trained using 10000 records of the CIFAR-10 dataset, using SGD with momentum and L2 regularization penalty 0.001.
>   - (b) trained using 10000 records of the Purchase100 dataset, using SGD with clipping norm = 2.0
>   - The AUC scores of the well-generalized models have been added to the respective dataset's AUC score tables. Scores for (a) can be found in the CIFAR10 AUC Scores table (Table 3 in the Appendix). Scores for (b) can be found in Configuration IV of the Purchase100 AUC Scores table (Table 1). The FPR-TPR curves of these configurations have been added in Figure 4 in the Appendix as well.
> - The attack performance for well-generalized models are lower than that for over-fitted models, which is consistent with previous results [a] that shows well-generalized models have lower information leakage.
> - However, we want to emphasize that the general comparison between the performance of different models persists. That is, as the attack exploits its dependence on specific models and specific data samples better, the attack performance improves. More specifically, the performance (AUC) of attack S $\approx$ attack P $<$ attack R $<$ attack D. This illustrates that by exploiting model-dependence and (or) sample-dependence in loss threshold-based membership inference attacks, we can obtain more efficient and (or) more powerful attacks.
>
> **Q2**
> > The comparison and difference between this work and prior work is vaguely discussed. No in detailed comparison with [1], which also formalized MIA with indistinguishability based framework. The hypothesis testing framework in this work is very similar to the threshold adversary in [1]. I hope authors could discuss the differences in detail. No comparison to white-box models (e.g. [2]). How would this compare to the proposed framework? Minor: related works should be discussed in the main text instead of supplementary materials.
>
> With a few exceptions, most of the prior works are based on Attack S, and a few remaining related works could be considered as variations of Attack R. We will consider adding a more detailed description of the attacks and a comparison with previous membership inference attacks.
>
>
> **Q3**
> > No connection made between this work and differential privacy, although the indistinguishability setup is highly related. It would be nice to see how the framework translates to privacy loss in DP.
>
> The notion of privacy loss in our framework is based on differential privacy, but the metric is not exactly the same as differential privacy. Please see our response to Q5 for reviewer XT1N for more details.
>
>
> **Q4**
> > Minor: column names overflowed in Table 4.
>
> We fixed this in the revised version.
>
>
> References:
> - [a] Jayaraman, Bargav, and David Evans. "Evaluating differentially private machine learning in practice." USENIX Security. 2019.

---

### Official Review · Reviewer_XT1N · 2021-11-03

**Correctness:** 2
**Technical Novelty And Significance:** 3
**Empirical Novelty And Significance:** 2
**Recommendation:** 5
**Confidence:** 4

**Main Review:**

Please list both the strengths and weaknesses of the paper. When discussing weaknesses, please provide concrete, actionable feedback on the paper.

On the positive side, the paper aims to make “auditing privacy” more modular. This goal is pursued by proposing multiple different attacks for membership inference. Then, many different factors, beyond just the success rate of the attacks are compared and reported with experiments.

On the weakness side: it is not really clear how having multiple attacks of specific form can give a full picture about whether or not an algorithm is private. There is of course the possibility that none of these attacks work on a learning algorithm and another attack works. That is why the only way to really “audit” privacy is to prove it. Alternatively, one might envision being able to prove that some class of attacks are “complete” in some sense, but I doubt that we are there yet, and certainly this paper does not provide any justification as to why their set of attacks are “complete”. In general, the paper’s plans are cryptic and each page has many ambiguous sentences with unclear goals. That is also the case for the introduction. The paper also does not justify clearly why they pick certain criteria and focus on them. For example, the bullets in page 3 are not clearly motivated/explained.

Other comments:

There are quite a few typos, plz make a pass. Example “confidentiality confidentiality” “explaining about the uncertainties” etc.

The introduction says “"Below, we first briefly review the related work in this domain, explain how different works measure privacy risk, and then…”
But then the paper moves on and explains the contributions of our work. I guess that is maybe due to the change of the paper for the submission.

The paper says “The notion of privacy underlying our framework is primarily based on differential privacy,  I would say computational DP”
Note that your definition is rather an “average case, computational, indistinguishability-based” variation of DP. That is because: you pick the data set and z at random, you work with efficient (poly time) attacks and your goal is just to distinguish 2 cases. Standard DP asks for more conditions in all three categories.

The first attack writes probabilities P(theta|D) as if we are aware of the learning algorithm. Can you please be more clear that you are basically *assuming* P(theta|D) to be what you wrote in Equation 1? In other words, you are assuming to work with a specific learner, because you sate the distribution of theta given D, no?

Page 3: “The population data used for constructing the attack algorithm, and evaluating the inference game, need to be similar, in distribution, to the training data”
I don’t get this (and many similar firm judgements). Why? I agree that natural algorithms might fall into this category, but simply stating it as a general rule needs proof (and I don’t think this is true in general).

“By violating this principle, we might overestimate the privacy loss”
I think this is wrong. The whole point of DP is that the adversary might have *arbitrary* auxiliary information. So, I see no reason why having a particular auxiliary information, and winning the security game, can be a misleading indication that the scheme was not DP; quite the contrary.

As another example of sentences that I found unclear:
“The adversary knows the underlying data distribution
What does that mean? Can I sample from it?

Maybe I am missing something here, but I don't think pi(z) should show up in Equation (4). Is not z independent of \theta and D in that case?

In your experiments of the main body: you say the data set (Purchase100) but what is the learner?


**Summary Of The Paper:**

Membership inference attacks are attacks that infer whether a given record is in the dataset of a given machine learning model or not. By definition, (the success of) such attacks is in contrast with differential privacy.

The paper proposes multiple attacks to infer membership in the data sets of machine learning models. The envisioned goal of the paper is that these attacks can provide a more complete picture about how private a machine learning algorithm is, when compared with other membership inference attacks. Hence, the ultimate goal is to provide “reports” that go beyond “just a number” to better audit the privacy of learning algorithms.


**Summary Of The Review:**


I agree with the general sentiment of the paper, that having a framework for auditing privacy could be helpful. But such efforts need a much more detailed and justified approach, arguing for “complete” attacks that at least “capture known attack techniques so far”. Otherwise, just focusing on 5 attacks might be dangerous as that might suggest that schemes are private while they are not. I also think that the paper needs to be much more clear in its criteria that it proposes (in addition to the success rate of the attacks) to be part of the “report” on privacy.

---

> ### Author Response · Authors · 2021-11-17
> **Response for reviewer XT1N (1/3)**
>
> **Q1**
> > ... it is not really clear how having multiple attacks of specific form can give a full picture about whether or not an algorithm is private. There is of course the possibility that none of these attacks work on a learning algorithm and another attack works. That is why the only way to really "audit" privacy is to prove it. Alternatively, one might envision being able to prove that some class of attacks are "complete" in some sense, but I doubt that we are there yet, and certainly this paper does not provide any justification as to why their set of attacks are "complete".
>
> - We first emphasize that privacy attacks are **not meant to certify** that a given algorithm is private, but rather to **test** whether it is not privacy-preserving. The attacks can reflect the lower-bound on the information leakage. The role of privacy attack for auditing privacy risk in machine learning, is similar to the role of cryptanalysis for cryptosystem (or similar to the role of penetration testing in security).
>
> - Secondly, the attacks presented in the paper are not meant to cover different types of leakage, but rather to help us design a more precise attack (Attack D). The attack D is the most powerful attack compared to all prior attacks.
>
> - Thirdly, our derivations of the new membership inference attacks are not random, but rather are all based on LR tests which can be formulated as comparing the loss of the target data on the target model with a loss threshold. We study the central problem that: using black-box access to the target model, how can we derive better loss threshold attacks by making the attack strategy model-dependent and (or) sample dependent? Our main conclusion is, by exploiting the dependence of loss threshold on specific models and (or) specific samples, the loss threshold-based attacker can become more efficient (Attack P) or achieve better attack performance (Attack D).
>
>
> **Q2**
> > The paper also does not justify clearly why they pick certain criteria and focus on them. For example, the bullets in page 3 are not clearly motivated/explained.
>
> - Firstly, we want to emphasize that it is not that we pick certain criteria, but rather we are studying all the criteria that affect the performance (accuracy, AUC) of the attack. The attack performance *depends* on the uncertainties in the process of producing a model. Thus, they need to be explicitly stated.
>
> - The bullet points in page 3 are the factors in the evaluation process that could affect the evaluated attack performance. We are interested in measuring the true leakage. Meanwhile, we want to eliminate the influence of other factors, such as attacker's prior for the evaluation data and training algorithm, on the attack performance. See the example that we give in response to Q7 for more details.
>
> - We understand that the confusion may be due to our writing. We will further clarify these criteria in the future.
>
>
> **Q3**
> > There are quite a few typos, plz make a pass. Example "confidentiality confidentiality" "explaining about the uncertainties" etc.
>
> Thank you for the comment, we address these typos in the revised version.
>
> **Q4**
> > The introduction says ""Below, we first briefly review the related work in this domain, explain how different works measure privacy risk, and then…" But then the paper moves on and explains the contributions of our work. I guess that is maybe due to the change of the paper for the submission.
>
> We fixed this inconsistency in the revised version.

---

> > ### Author Response · Authors · 2021-11-17
> > **Response for reviewer XT1N (2/3)**
> >
> > **Q5**
> > > The paper says "The notion of privacy underlying our framework is primarily based on differential privacy, I would say computational DP" Note that your definition is rather an "average case, computational, indistinguishability-based" variation of DP. That is because: you pick the data set and z at random, you work with efficient (poly time) attacks and your goal is just to distinguish 2 cases. Standard DP asks for more conditions in all three categories.
> >
> > - To clarify our statement in the paper: we mean that the information leakage with respect to membership inference attacks is essentially the same *type* of information leakage which is the foundation of differential privacy metric. We understand that DP is a bound on this leakage over all possible neighboring datasets and randomnesses of the algorithm. But, it is the worst case of the same notion of leakage.
> >
> > - We acknowledge that the leakage could be measured as an average case metric. However, the metric itself is not necessarily an "average case". In sections 3.1-3.4, we are also trying to derive membership inference attacks for one specific model and (or) specific target data point. In this way, we are NOT defining a new privacy notion, but rather, pushing for the limit of using model-dependent and (or) sample dependent membership inference attacks to measure the more accurate privacy leakage based on standard DP. This also helps us to identify vulnerable data points and vulnerable models. Please see our response to reviewer SgO6, Q2 for more details.
> >
> >
> > **Q6**
> > > The first attack writes probabilities P(theta|D) as if we are aware of the learning algorithm. Can you please be more clear that you are basically assuming P(theta|D) to be what you wrote in Equation 1? In other words, you are assuming to work with a specific learner, because you sate the distribution of theta given D, no?
> >
> >
> > No, we are not restricting ourselves to a specific learner.
> >
> > - We explain the origin of this posterior distribution assumption in Bayesian learning, its applicability to different learners such as SGD, Bayesian posterior sampling, and MAP (max a priori) learners, and its usage in previous works [a] of membership inference attacks more clearly in the revised paper.
> >
> > - We emphasize that this posterior assumption only helps us explain why a loss threshold-based attack strategy is a reasonable approach (used and justified in many prior works). It is very important to note that, however, our computation of attack threshold in Section 3, holds for general loss threshold-based membership inference attacks. In Section 4, we also extend the evaluations of the attacks to various learning algorithms, such as SGD with clipping, momentum and regularization, on deep neural networks.
> >
> >
> > **Q7**
> > > Page 3: "The population data used for constructing the attack algorithm, and evaluating the inference game, need to be similar, in distribution, to the training data" I don't get this (and many similar firm judgements). Why? I agree that natural algorithms might fall into this category, but simply stating it as a general rule needs proof (and I don't think this is true in general).
> >
> > - Yes, to correctly audit the information leakage with attack performance, the attack evaluation must be done on data that are similar, in distribution, to the training data. This is because population data used for evaluating attacks affect its performance drastically, even when we are measuring the same membership inference attack strategy, on the same target model and target sample data.
> >   - As an example, if during evaluation of the attacks, we only use rare data (that the attacker knows a priori to be very unlikely to appear in the training set), then a naive attacker that outputs non-member for every data sample would achieve high performance (100% attack accuracy and AUC score of one). However, this high attack performance does not imply anything about the information leakage in the target model, about a target training data sample. Rather, this high attack performance only captures the attacker's knowledge of the population data used in evaluation.
> > - To measure the leakage from the model, we don't want other factors (that do not necessarily imply high information leakage) contributing to the high performance of the attack. Therefore, we need the data distributions during training and attack evaluation to be similar, so as to eliminate these factors (that are unrelated to the actual information leakage), and ensure that our measured attack performance truthfully reflects the information leakage.

---

> > > ### Author Response · Authors · 2021-11-17
> > > **Response for reviewer XT1N (3/3)**
> > >
> > > **Q8**
> > > > "By violating this principle, we might overestimate the privacy loss" I think this is wrong. The whole point of DP is that the adversary might have arbitrary auxiliary information. So, I see no reason why having a particular auxiliary information, and winning the security game, can be a misleading indication that the scheme was not DP; quite the contrary.
> > >
> > > - Any inference attack depends on prior knowledge and information leakage. DP bounds the information leakage. The famous example in the "algorithmic foundations of DP" is that DP can only hide the additional privacy risks of including data in the input dataset, with respect to inference attacks. It cannot bound the absolute privacy value: if I know Alice is 1 inch taller than the average Russian, by releasing an average over a dataset that does not include Alice's data (thus does not leak about it), you could precisely infer Alice's hight. Auxiliary information does contribute to the attacker's performance even against DP models/statistics. What DP mitigates is the "additional" information attack performance due to including one record in the dataset.
> > >
> > > - We want to remove (reduce as much as possible) the dependency of attack performance on all factors except the information leakage of the model about its training data. Thus we need to make that assumption in the process of auditing models.
> > >
> > > - So, membership inference attack evaluations generally require this assumption, that evaluation data are taken i.i.d. from an overall population data distribution that is similar to the training data for the target models [b][c]. Otherwise, the attack performance (accuracy and AUC over random target data samples) depends on population prior distribution, thus may not accurately capture privacy leakage and powerfulness of the attack. (It could just be the case that the attacks has a relatively accurate prior, see a detailed example in our response to Q7.) We refer to the [d] for more insightful discussions.
> > >
> > > - What's more, our attack evaluation metrics (FPR and TPR) are model-specific, instead of algorithm-specific in the standard hypothesis formulation of differential privacy. More specifically, we measure the FPR over random samples of target data from the population distribution. Meanwhile, the FPR in the common hypothesis testing framework of DP is measured over random sample models trained using the same algorithm on the same training dataset.
> > >
> > >
> > > **Q9**
> > > > As another example of sentences that I found unclear: "The adversary knows the underlying data distribution What does that mean? Can I sample from it?
> > >
> > > Yes, the adversary can sample from the underlying data distribution, to construct population datasets.
> > >
> > > **Q10**
> > > > Maybe I am missing something here, but I don't think pi(z) should show up in Equation (4). Is not z independent of \theta and D in that case?
> > >
> > > - This $\pi(z)$ in Equation (4) occurs because z is taken from the population distribution, where the population distribution is denoted as $\pi(z)$, independent from the target dataset $D$.
> > >
> > >
> > > **Q11**
> > >
> > > > In your experiments of the main body: you say the data set (Purchase100) but what is the learner?
> > >
> > > The learner for Purchase100 experiments consists of:
> > > 1. A 4 layer MLP network with layer units = [512, 256, 128, 64].
> > > 2. The optimization algorithm is stochastic gradient descent. Configurations 1b, 2b are trained using L2 regularization with regularization penalty 0.01. Configuration 4 is trained with a gradient clipping norm of 2.0.
> > >
> > >
> > > References:
> > > - [a] Sablayrolles, Alexandre, et al. "White-box vs black-box: Bayes optimal strategies for membership inference." ICML 2019.
> > > - [b] Jayaraman, Bargav, and David Evans. "Evaluating differentially private machine learning in practice." USENIX Security. 2019.
> > > - [c] Yeom, Samuel, et al. "Privacy risk in machine learning: Analyzing the connection to overfitting." 2018 IEEE 31st Computer Security Foundations Symposium (CSF). IEEE, 2018.
> > > - [d] Humphries, Thomas, et al. "Differentially Private Learning Does Not Bound Membership Inference." arXiv preprint arXiv:2010.12112 (2020).

---

> > ### Comment · Reviewer_XT1N · 2021-11-22
> > **Thanks**
> >
> > I find some of the points that were made in the response by the authors useful. Particularly, the comments to the first point (e.g., those about cryptanalysis). I think the paper would benefit from such clarifications (in general about what the contribution is exactly, and specifically with regard to the limitations of the membership inference as a way of ensuring privacy). I'd suggest also revising some of the sentences that from the abstract ; e.g., "Our algorithms capture a very precise approximation of privacy loss in models, and can be used as a tool to perform an accurate and informed estimation of privacy risk in machine learning models."
> > This sentence can very easily be (mis)understood as : "if our attacks do not succeed, the model is private".
> >
> > All in all, my evaluation of the paper improved after the response (hence I increase my score), but I think the paper can really benefit from a full pass to further clarify the main contributions, and (since the contribution is also about attacks) more detailed comparison with previous attacks.

---

> > > ### Author Response · Authors · 2021-11-23
> > > **Response to Reviewer XT1N**
> > >
> > > Thanks for the feedback.
> > > > "Our algorithms capture a very precise approximation of privacy loss in models, and can be used as a tool to perform an accurate and informed estimation of privacy risk in machine learning models." This sentence can very easily be (mis)understood as : "if our attacks do not succeed, the model is private".
> > >
> > > Thanks for pointing out this confusion due to our writing. Here by "precise approximation of privacy loss" we mean we measure model-specific and sample-specific privacy loss, instead of general privacy loss of the algorithm on a data population.
> > >
> > > We will further clarify these confusions regarding using privacy attacks for auditing privacy loss, our main contributions, and comparison with previous attacks in future revisions of this paper. Thanks.

---

### Official Review · Reviewer_Sgo6 · 2021-11-03

**Correctness:** 3
**Technical Novelty And Significance:** 2
**Empirical Novelty And Significance:** 2
**Recommendation:** 3
**Confidence:** 4

**Main Review:**

** Strengths **
- The investigated problem is relevant and the paper is well written;
- The notation is convient enough and all important concepts are well introduced;
- The central idea remains interesting but it is not novel enough.

** Weaknesses **
- The contributions are not clearly stated in the introduction section. Therefore, it appears difficult to identify which are the main contributions in this paper. According the introduction section, the main contribution is the proposed framework via the connection with binary hypothesis testing (i.e., the tradeoff between Type I and Type II errors) seem to be the main contribution.
- The main misconception underlaying this work is the lack of novelty. The proposed framework via the connection with binary hypothesis testing is not novel and has been already proposed in a series of works (e.g., among others, see https://arxiv.org/abs/2105.03875).
- All results in the paper rely strongly on the assumption introduced in eq. (1) which is indeed to characterize the outcome of the algorithm obtained via SGD training. However, this has not been properly justified to be considered as being strong enough to validate a possible privacy audit of ML  algorithms. The authors do not provide any valide proof or formal justification for this assumption.
- Another central problem is the lack of numerical results and in particular, comparison with competing methods for building attacks in the literature. In addition the tradeoffs between Type I and Type II error are not always reported. On the other hand, there is very little data sets and relevant architectures to validate the results.


**Summary Of The Paper:**

The focus of this paper is on membership inference attacks. In particular, the paper aims at providing a framework that can help to access how different factors beyond information leakage from the model affect the performance of membership inference attacks, and how to design attacks that cancel out the effects off other factors. More specifically, a framework for understanding the relationship between, success of membership inference attacks and information leakage is introduced. Experiments using well-known datasets on visual tasks are provided to evaluate the performance  of the proposed attacks(in terms of AUC scores. Various frameworks are discussed in Section 3.

**Summary Of The Review:**

Given the above comments, I believe that the paper lacks of technical novelty, numerical results and relevant comparisons with several existent methods in the literature. To conclude, I believe that the work contains some valuable ideas which deserve to be further investigated (both theoretically and practically)  but in its current state, the contributions and the presentation of the results are too short in terms of exceptions to be accepted to be published in ICLR conference.

---

> ### Author Response · Authors · 2021-11-17
> **Response to reviewer SgO6 (1/2)**
>
> **Q1**
> > The contributions are not clearly stated in the introduction section. Therefore, it appears difficult to identify which are the main contributions in this paper. According the introduction section, the main contribution is the proposed framework via the connection with binary hypothesis testing (i.e., the tradeoff between Type I and Type II errors) seem to be the main contribution.
>
> - Our major contribution is NOT to propose a unifying hypothesis testing framework for membership inference attacks, but rather to **design new membership inference attacks** that are model-dependent and sample-dependent by decomposing the null hypothesis in our hypothesis testing framework. The framework is there to enable us to systematically compare different approaches for designing attacks. The prior work on ML attacks is primarily based on Attack S. Within this framework we can motivate why such prior attacks are insufficient for analyzing privacy risks of models. Not only do we justify the reasons behind this new attack, we also compare the attack performance (TPR-FPR curve and AUC score) and computation cost of our new attacks (notably Attack D[istillation]) with prior work.
>
> - We would like to highlight that our proposed new attack P is more efficient than the shadow model attack (as it does not require training new shadow models) while achieving roughly the same (or slightly better) attack performance (in Figure 2). Moreover, our new attack D via distillation is powerful and achieves a higher AUC score (in Figure 2) than shadow model attacks.
>
> **Q2**
> > The main misconception underlaying this work is the lack of novelty. The proposed framework via the connection with binary hypothesis testing is not novel and has been already proposed in a series of works (e.g., among others, see https://arxiv.org/abs/2105.03875).
>
>
> - Yes, we are aware of various previous works that formalize average-case membership inference attacks inside the binary hypothesis testing framework.
> - However, our **main novelty** is not the derivation of this average case test, but rather, a general methodology of **exploiting dependence** of attacks on specific target **model** and (or) target data **sample**, to **design new privacy attacks** (attack P, D) that pushes for **better attack performance**.
>   - By fine-grained decomposition of the null hypothesis based on specific target model and (or) target data sample, we design new membership inference attacks that are either more efficient (attack P that does not need training new shadow models) or more powerful (attack D that are more powerful, i.e. has higher AUC).
>   - Moreover, by decomposing the null hypothesis at the label level (or sample level), we can also recover previous successful baseline attacks - shadow (or reference) model attack following this general methodology.
> - Besides designing new attacks, we also use the right evaluation metrics (model-specific FPR and TPR) for membership inference hypothesis testing that help us to compare the performance of different membership inference attacks, as well as to identify vulnerable models and records.
>   - That is, we audit the privacy leakage of one **specific target model** and **specific data samples**, rather than analyzing the average-case information leakage of any instance of the training algorithm on any data.
>
> **Q3**
> > All results in the paper rely strongly on the assumption introduced in eq. (1) which is indeed to characterize the outcome of the algorithm obtained via SGD training. However, this has not been properly justified to be considered as being strong enough to validate a possible privacy audit of ML algorithms. The authors do not provide any valide proof or formal justification for this assumption.
>
> - We explain the origin of this posterior distribution assumption in Bayesian learning, its applicability to different learners such as SGD, Bayesian posterior sampling, and MAP (max a priori) learners, and its usage in previous works [a] of membership inference attacks more clearly in the revised paper.
> - We emphasize that this posterior assumption only helps us explain why a loss threshold-based attack strategy is a reasonable approach (used and justified in many prior works). It is very important to note that, however, in computation of attack threshold in Section 3, hold for general loss threshold-based membership inference attacks. In Section 4, we also extend the evaluations of the attacks to various learning algorithms, such as SGD with clipping, momentum and regularization, on deep neural networks.

---

> > ### Author Response · Authors · 2021-11-17
> > **# Response to reviewer SgO6 (2/2)**
> >
> > **Q4**
> >
> > > Another central problem is the lack of numerical results and in particular, comparison with competing methods for building attacks in the literature. In addition the tradeoffs between Type I and Type II error are not always reported. On the other hand, there is very little data sets and relevant architectures to validate the results.
> >
> > With a few exceptions, most of the prior works are based on Attack S, and a few remaining related works could be considered as variations of Attack R. We will consider adding a more detailed description of the attacks and a comparison with previous membership inference attacks, as well as adding more experiments results on different datasets and model architectures.
> >
> >
> > References:
> > - [a] Sablayrolles, Alexandre, et al. "White-box vs black-box: Bayes optimal strategies for membership inference." ICML 2019.

---

> > > ### Comment · Reviewer_Sgo6 · 2021-11-28
> > > **Acknowledgement of authors' response**
> > >
> > > I have read the rebuttal and the revised paper. However, I do not think that my concern raised in Q3 has been properly addressed (e.g., formally addressed). In addition, I believe that this work  lacks of enough novelty since most of the proposed concepts were already in the literature.

---

> > > > ### Author Response · Authors · 2021-11-29
> > > > **Response to reviewer Sgo6**
> > > >
> > > > > I do not think that my concern raised in Q3 has been properly addressed (e.g., formally addressed).
> > > >
> > > > The concern for Q3 is addressed in the revised paper, during the construction of the LRT strategy, above equation (6). We explained the applicability of this Bayesian posterior sampling equation in (Bayesian) machine learning algorithms.
> > > > - We believe theoretical justification of how and why certain randomized learning algorithms can be viewed as this Bayesian posterior sampling, however, is not a contribution of this paper, and it is extensively studied in various works that focus on Bayesian approaches such as SGLD [1]. Meanwhile, our paper focuses exclusively on constructing membership inference attacks.
> > > > - Moreover, in the construction of attacks and experiments in Sections 3 and 4, we do not restrict ourselves to algorithms that strictly satisfy the conditions for this Bayesian posterior sampling equation to hold. More specifically, our experiment involves learning algorithms based on stochastic gradient descent on deep neural networks. (See experiment setup in Appendix A.1 for more details.)
> > > > - We will consider adding more references to various formal justification of this posterior equation for Bayesian learning algorithms, in future versions of the paper.
> > > >
> > > >
> > > > > In addition, I believe that this work lacks of enough novelty since most of the proposed concepts were already in the literature.
> > > >
> > > > We include the shadow model attack and the reference model attack (proposed in previous work) in our paper mainly to explain them under the perspective of attacker uncertainty for information leakage. We **use these prior works as baselines** to show what uncertainties an attack could characterize, and explain why the different captured uncertainty affects their ability to capture model-specific and sample-specific information leakage.
> > > > - For example, shadow model-based attacks use the same loss threshold for different target samples (in the same class), which is too general to account for specific samples. A sample could have a larger loss than that constant threshold, while it is a member of the training dataset, simply because it is a hard-to-learn example (that tend to have a high loss on all models trained with it).
> > > > - Motivated by this example, we construct new attacks P and D that captures model-specific (and sample-specific) information leakage. We show how our newly constructed attack D offers more precise information leakage quantification (by reducing the attacker uncertainty to the specific target model and target sample). We also found that the model-specific new attack P achieves similar performance as the shadow model attack while significantly less computation cost.
> > > >
> > > > [1] Zhang, Y., Liang, P., & Charikar, M. (2017, June). A hitting time analysis of stochastic gradient langevin dynamics. In Conference on Learning Theory (pp. 1980-2022). PMLR.

---

### Decision · Program_Chairs · 2022-01-20

**Decision:**

Reject

**Comment:**

This work provides a formal framework for discussing membership inference attacks (MIA). It then examines existing attacks and proposes some new ones. The attacks are evaluated on several datasets.  The framework mostly formalizes the types of information and error types that an attack may use and is presented as the main contribution of this work. However the presented formalizations do not appear to contribute significantly beyond the existing work on MIAs. The new attacks may be of interest and, according to the presented experiments, (mildly) improve on some of the existing MIAs. At the same time, as presented, the discussion of the the benefits of the new attacks is relatively short and reviewers did not find the results to be sufficiently convincing. Therefore I cannot recommend acceptance for this work in its current form.